# Application of Machine Learning Algorithms to Describe the Characteristics of Dairy Sheep Lactation Curves

**DOI:** 10.3390/ani13172772

**Published:** 2023-08-31

**Authors:** Lilian Guevara, Félix Castro-Espinoza, Alberto Magno Fernandes, Mohammed Benaouda, Alfonso Longinos Muñoz-Benítez, Oscar Enrique del Razo-Rodríguez, Armando Peláez-Acero, Juan Carlos Angeles-Hernandez

**Affiliations:** 1Centro de Ciências e Tecnologias Agropecuárias, Universidade Estadual do Norte Fluminense, Campos dos Goytacazes 28013-620, Brazil; lilian.mvz@gmail.com (L.G.); alberto@uenf.br (A.M.F.); 2Instituto de Ciencias Básicas e Ingeniería, Universidad Autónoma del Estado de Hidalgo, Pachuca 42184, Mexico; fcastro@uaeh.edu.mx; 3Institut Agro Dijon, 26 Bd Dr Petitjean, 21079 Dijon, France; mohammed.ben-aouda@agrosupdijon.fr; 4Instituto de Ciencias Agropecuarias, Universidad Autónoma del Estado de Hidalgo, Tulancingo de Bravo 43600, Mexico; alfonso_munoz@uaeh.edu.mx (A.L.M.-B.); oscare@uaeh.edu.mx (O.E.d.R.-R.); pelaeza@uaeh.edu.mx (A.P.-A.)

**Keywords:** artificial intelligence, milk production, milk recording, sheep

## Abstract

**Simple Summary:**

An accurate estimation of the characteristics lactation curves is required to optimize sheep milk production. The adjustment of the lactation curve is traditionally been performed using mathematical models through linear and non-linear regression. However, these analytical tools have several limitations, mainly related to the non-linear pattern of the lactation curve. Machine learning algorithms have been used successfully to model and predict complex biological processes. In the current study, we evaluated the ability of seven machine learning algorithms, including linear and non-linear regression, to estimate total milk yield, peak yield, and time to peak yield of dairy sheep lactations. In addition, the estimates provided by machine learning algorithms were compared with the Wood model and the observed values. All algorithms tested showed good estimates, with the SMOreg algorithm showing the best performance. Furthermore, our results indicated that adequate estimates can be obtained with only five milk records. Therefore, machine learning algorithms are an option to correctly predict the characteristics of the lactation curve of dairy sheep, optimizing the use of available data.

**Abstract:**

In recent years, machine learning (ML) algorithms have emerged as powerful tools for predicting and modeling complex data. Therefore, the aim of this study was to evaluate the prediction ability of different ML algorithms and a traditional empirical model to estimate the parameters of lactation curves. A total of 1186 monthly records from 156 sheep lactations were used. The model development process involved training and testing models using ML algorithms. In addition to these algorithms, lactation curves were also fitted using the Wood model. The goodness of fit was assessed using correlation coefficient (r), mean absolute error (MAE), root mean square error (RMSE), relative absolute error (RAE), and relative root mean square error (RRSE). SMOreg was the algorithm with the best estimates of the characteristics of the sheep lactation curve, with higher values of r compared to the Wood model (0.96 vs. 0.68) for the total milk yield. The results of the current study showed that ML algorithms are able to adequately predict the characteristics of the lactation curve, using a relatively small number of input data. Some ML algorithms provide an interpretable architecture, which is useful for decision-making at the farm level to maximize the use of available information.

## 1. Introduction

The study of the mathematical properties of the lactation curve provides a summary of the evolution of milk production, which is a multifactorial process determined by the interaction between the environment and the biological efficiency of the animal [1]. The analysis of the lactation curve is a valuable tool in research and herd management, as it is useful for estimating total production during lactation with incomplete records. This is essential information for farmers to make management decisions, implement and evaluate genetic improvement programs, monitor health, forecast feed needs, and address economic and management aspects [2].

Mathematically, the shape of the lactation curve is defined by its characterizing parameters, such as total milk yield, peak yield, and time to reach the peak yield [2]. Traditional statistical techniques such as linear regression, non-linear regression, and random regression are used to calculate the parameters of the lactation curve using empirical and mechanistic approaches [3]. In addition, in recent years, machine learning (ML) algorithms have emerged as powerful tools for predicting and modeling complex data. These algorithms have the ability to learn patterns and hidden relationships in large data sets.

For dairy cattle, ML has helped to predict mastitis, to detect oestrus, and to estimate milk production [3,4,5,6]. In the case of buffaloes, the implementation of ML algorithms produced a correct estimation of the peak of milk production [7]. More recently, artificial neural networks (ANN) have been used to estimate milk production based on udder measurements in sheep, showing a better fitting performance compared to multiple regression [8]. Due to their capacity to learn from complex relationships between data and produce accurate predictions, the ML algorithms are a promising analytical method for the estimation of milk production in sheep and for a more accurate description of the lactation curve. Therefore, the aim of the current study was to evaluate the goodness of fit of different ML algorithms and a traditional empirical model for the estimation of the parameters of the lactation curve.

## 2. Materials and Methods

### 2.1. Database

A total of 1186 monthly records were used according to the A4 method proposed by ICAR [9] as the standard method for recording milk production in dairy sheep. These records were obtained from 156 multiparous (second and third lambing) sheep lactations from a commercial farm located in the Querétaro region, Mexico, with an average annual temperature of 17.3 °C and an average annual rainfall of 485 mm. Lactations with a minimum of five monthly records were selected, resulting in a database of 119 lactations. We analyzed the lactation curves of dairy crossbred ewes from the following breeds: East Friesian, Pelibuey, Suffolk, and Black Belly. The average lactation length was 237.4 days, with a total milk yield (TMY) of 102.0 l, peak yield (PY) of 0.97 l, and a time-to-peak yield (TPY) of 30 days.

A descriptive analysis and an outlier screening were performed, which resulted in the exclusion of lactations with TMY greater than 182 l (*n* = 8) and TPY greater than 100 days (*n* = 6). The current study was performed using a final database of 105 lactations. Actual total milk yield was calculated from monthly records of milk production using the centering day method or Fleischmann’s method [10]. PY and TPY were determined by identifying the highest values on each lactation curve.

### 2.2. Model Formulation

The following lactating traits were defined as input attributes: (1) first day of milk production recording; (2) lactation duration (in days); (3) milk yield at first monthly recording (MP1; l/d); (4) milk yield at second monthly recording (MP2; l/d); (5) milk yield at third monthly recording (MP3; l/d); (6) milk yield at fourth monthly recording (MP4; l/d); (7) milk yield at fifth monthly recording (MP5; l/d). The output attributes corresponded to the following lactation characteristics: TMY, PY, and TPY. Formulating the model involves mapping the input attributes (1 to 7) to produce the values of the output attributes.

The model development process involved training and testing models using ML algorithms independently for each output variable. During training, predictive models were constructed using the input and output datasets. The Waikato Environment for Knowledge Analysis (WEKA, version 3.8.6) was adopted as the standard interface to compare different data mining algorithms and determine the best analytical approach. WEKA is a Java-based data mining software available as open source. The database was transformed into the data format compatible with WEKA (attribute file format; .arff) in order to run the algorithms. Therefore, the final dataset consisted of 7 independent attributes and one dependent attribute (TMY, PY, or TPY), depending on the characteristic to be determined. The following seven ML algorithms were then selected from the 28 available in WEKA based on the results of the root mean square error:Gaussian Processes (GP): is a powerful and flexible tool for modeling and prediction in regression and probabilistic classification. GP Regression is a probabilistic model that defines a distribution over functions that implement Gaussian processes for regression without hyper-parameter tuning. To facilitate the selection of an appropriate noise level, GP applies normalization/standardization. GP regression also allows the description of non-linear relationships between input and response variables, as well as data uncertainty [11].Linear Regression (LR): is a popular and fundamental statistical modeling technique used to predict a continuous output variable based on one or more input features. It is a supervised learning algorithm that aims to determine the best linear relationship between the input features, and the output variable, where the weights are calculated from the training data and the Akaike Information Criterion, is used for model selection [12].Multi-layer Perceptron (MP): is a non-linear information algorithm inspired by the biological nervous system. This type of artificial neural network is based on interconnected elementary processing devices called neurons. The network starts from the input information through one or more hidden layers to the input layers [8].Sequential Minimal Optimization Regression (SMOreg): SMO is an algorithm for efficiently solving optimization problems that arise when training a support vector machine. SMO helps to deal with the problem of quadratic programming associated with the optimization of the analytical, eliminating the need to use an iterative quadratic programming optimizer as part of the algorithm. Shevade et al. [13] proposed an iterative algorithm based on SMO to deal with regression problems.M5 Rules (M5): generates a decision list for regression problems using separate-and-conquer. In each iteration, it builds a model tree using M5 and makes the “best” leaf into a rule [12].M5 model tree (M5P): is a decision tree learner for regression tasks used to predict values of numerical response variables. It is a binary decision tree with linear regression functions at the terminal (leaf) nodes that can estimate continuous numerical attributes. The construction of the M5 model tree involves two steps. The first step involves the use of splitting criterion to create a decision tree, and the second step involves pruning the overgrown tree and replacing the sub-trees with linear regression functions [14].Random Forest (RF): is an ensemble learning method that combines multiple decision trees to make predictions. It is a versatile and powerful algorithm that can be used for both regression and classification tasks. RF for regression is formed by growing trees depending on a random vector that takes numerical values as opposed to class labels. The output values are numerical, and it is assumed that the training data set is independently drawn from the distribution of the random vector [15].

Each model was trained and then validated using 10-fold cross-validation. The original dataset was divided into ten equal partitions, and in each iteration, the data were split into training and testing sets, ensuring that each instance was used once for testing. 

### 2.3. Mathematical Modeling

In addition to these seven algorithms, lactation curves were also fitted using the incomplete gamma empirical model proposed by Wood [16]:*Y_t_* = *at^b^e*^−*ct*^,(1)
where *Y* is the milk production (l) at time *t*, *a* is the production at the beginning of lactation, *b* represents the ascending phase, and *c* describes the descending phase of the lactation curve. The parameters of the Wood model (*a*, *b*, and *c*) were estimated through the iterative process of non-linear curve fitting in regression analysis using the “nlsLM” function from the “minpack.lm” package [17] in the R statistical computing environment (version 4.3.0; R Core Team, 2020) with all the monthly records from 105 lactating sheep. Based on the estimated parameters, the following lactation curve characteristics were calculated: TMY, PY, and TPY.

### 2.4. Performance Evaluation Criteria

The goodness of fit of the models was evaluated using the following metrics:

Coefficient of Correlation (r): This metric measures the strength and direction of the linear relationship between the predicted and the actual values. A value close to 1 indicates a strong positive correlation, while a value close to −1 indicates a strong negative correlation.Mean Absolute Error (MAE): This represents the average absolute difference between the predicted values and the actual values. It measures the average magnitude of the errors without considering their direction. It is represented by:

MAE = (1/n) ∗ Σ|y_i_ − x_i_|,(2)
where y_i_ and x_i_ are the observed and estimated values, respectively, and n is the total number of observations.

3.Root Mean Square Error (RMSE): This is calculated by taking the square root for the average of the squared differences between the predicted values and the actual values. The RMSE provides a measure of the overall magnitude of the errors, giving more weight to larger errors.

RMSE = √((1/n) ∗ Σ(y_i_ − x_i_)^2^),(3)
where y_i_ and x_i_ are the observed and estimated values, respectively, and n is the total number of observations.

4.Relative Absolute Error (RAE): This metric is the ratio of the MAE to the mean of the actual values. It represents the average absolute difference between the predicted values and the actual values relative to the scale of the actual values.

RAE = (Σ|y_i_ − x_i_|)/(Σ|ȳ − x_i_|),(4)
where y_i_ and x_i_ are the observed and estimated values, respectively, and ȳ is the mean of the observed values.

5.Relative Root Mean Square Error (RRSE): Similar to RAE, RRSE is the ratio of the RMSE to the mean of the actual values. It represents the average magnitude of the errors relative to the scale of the actual values.

RRSE = √((Σ(y_i_ − x_i_)^2^)/(Σ(ȳ − x_i_)^2^)),(5)
where y_i_ and x_i_ are the observed and estimated values, respectively, and ȳ is the mean of the actual values.

Finally, the observed (*Y*) and the estimated values (*X*) from the Wood model and the ML algorithms were fitted using a linear regression model. The regression line and the predicted values were presented graphically in a scatter plot of predicted vs. observed values. In these scatterplots, the ordinate and abscissa have the same scale, and a 45-degree line has been drawn to facilitate their interpretation. The accuracy of the estimates is represented by the distance of a given point from the 45-degree line.

## 3. Results

In the current study, we evaluated the ability of ML algorithms to estimate lactation curve characteristics and found a better fit for all ML algorithms than the Wood model. The goodness of fit criteria for the Wood model and seven ML algorithms applied to sheep lactation curves are shown in Table 1. The estimated parameters for the Wood model are presented in the following equation:*Y* = (4.006)*t*^(0.055) ^exp^−(0.006)*t*^,(6)
where parameter *a* had a value of 4.006 l, corresponding to average milk production at the beginning of lactation. Parameters *b* (0.055) and *c* (0.006) represent the ascending and descending phases of the lactation curve. The values of parameters *b* and *c* > 0 indicate the probable presence of atypical curves.

Based on the coefficient of correlation values, the Wood model showed moderate goodness of fit (r = 0.68). RMSE represents the mean of residuals for each mathematical function on the original scale. According to our results, the Wood model had an average error of 41.9 l in estimating TMY. Regarding the ML algorithms, they all showed lower RMSE values compared to the Wood model (<34.4 l), where the SMOreg algorithm obtained the best TMY prediction with an average error of 11.0 l. Moreover, SMOreg obtained the best estimates for PY and TPY based on all goodness of fit criteria, such as r (0.80 and 0.69) and RMSE (0.2 l and 17.6 d). However, it is important to point out that LR, M5, and M5P goodness of fit was similar to the one shown by SMOreg. Figure 1 shows the relationship between the estimate and the actual values of TMY. These plots show that the magnitude of the residuals to TMY estimates are significantly lower for the ML algorithms compared to the Wood model, confirming their better predictive performance.

Table 2 shows the mean and significance test of the actual and estimated values of TMY, PY, and TPY using the Wood model and the best ML algorithm (SMOreg). There was a significant difference between the Wood model and SMOreg (*p* = 0.03) for TMY estimates. The SMOreg algorithm showed a better estimation of TMY with a difference of only 0.82 l compared to the actual value, which is considerably lower than the estimation error shown by the Wood model (14.1 l). In addition, as can be seen in Figure 2, the TMY was considerably overestimated by the Wood model in some lactations.

A trend (*p* = 0.07) was identified by comparing the actual values of PY and TPY with the estimates of the Wood model and the SMOreg algorithm. Once again, SMOreg showed a better estimate of PY than the Wood model (0.93 vs. 4.29 l), which overestimated TPY by 341.4%. Both estimation strategies underestimated the time of maximum production. The actual TPY was reached at 30 days postpartum, a value underestimated by seven days for the Wood model and five days for SMOreg.

## 4. Discussion

Our results show that all ML algorithms have a better predictive ability of lactation curve parameters than the Wood model. The Wood model is the most widely used mathematical function to describe the lactation curve, probably due to its simple mathematical structure and partial biological interpretation of its parameters [2]. However, as shown by the results of the current study, the Wood model is less accurate in estimating milk production in early lactation, PY and TYP [18], which could be related to the higher presence of atypical curves (without a lactation peak). Angeles-Hernandez et al. [18] indicated that 52.06% of sheep milk curves in Mexico had atypical shapes, which affected the fit of the Wood model due to significant variations in parameter *b*. The value of the *b* parameter in the Wood model is critical because it controls the degree of curvature in the milk curve, which affects the accuracy of estimating TMY, PY, and TPY [19]. 

In relation to the analytical procedure for fitting the lactation curve, the Wood model, like most lactation models, uses linear and non-linear regression for estimating its parameters and calculating lactation curve characteristics. [20]. The main advantage of these analytical approaches is the simplicity of implementing and interpreting their parameters [21]. However, linear regression methods have shown limited flexibility and poor predictive performance when the relationship between inputs and outputs cannot be reasonably established by linear function [22]. On the other hand, non-linear regression is more flexible and accurate for lactation curve fitting than linear regression [22], but they might be more sensitive to noise, outliers, or multicollinearity in the data than other ML algorithms. Therefore, the use of new analytical approaches, as proposed in the current study, is crucial to obtain better estimates of lactation curve characteristics. At the farm and research level, the accurate estimation of TMY, PY, and TPY allows the calculation of feed requirements, the costs associated with nutritional management, the evaluation of genetic potential, and the prevention of metabolic disorders [2,18].

In this sense, the proposed ML algorithms had a much better performance for the estimation of TMY, PY, and TPY of dairy sheep in comparison with the Wood model. However, there were some differences in the goodness of fit between the algorithms tested. The techniques used to generate the output attribute may explain the differences in goodness of fit between algorithms. In our study, the SMOreg algorithm showed the best performance for the estimation of TMY, PY, and TPY, which can be related to the fact that SMOreg aims to find the best regression function by projecting into a high-dimensional space where linear optimization techniques are applied and then projecting the resulting regression function back into the original low-dimensional feature space of the actual variables [23]. Additionally, this algorithm outperformed the others at predicting outcomes from databases containing noise and uncertain attributes due to its ability to normalize and separate data using a hyperplane as a decision boundary [24,25]. This is useful in biological problems such as milk production, where noisy data scenarios are common [26]. In the same line, our results are in agreement with Nguyen et al. [27], who indicated that support vector machine (SVM), the algorithm category in which SMOreg falls, is the most efficient method for milk production prediction in terms of accuracy and computational cost compared to multiple linear regression, MP and RF algorithms.

GP regression has been reported to be an efficient non-parametric tool for the development of prediction models. Compared to the Wood model, the GP algorithm showed a better prediction of PY and TPY in the current study. The improvement in estimating lactation curve characteristics may be related to the fact that GP requires no predefined fitting function specification and, therefore, has a general ability to adequately estimate all types of non-linear data [11], in our case, different lactation curve shapes. Studies reporting GP use in animal science are limited [28,29]. Baiz et al. [28] reported a higher accuracy (r^2^) and precision (RMSE) for the GP regression model in comparison with multiple regression models for the estimation of the energy content of maize for poultry. These authors pointed out that the superior performance of the GP is related to its higher degree of freedom and flexibility. Another advantage of GP is its ability to estimate an interpretable uncertainty. This is useful in the inference process to know the standard deviations of the predictions [29]. Compared to other ML algorithms, such as artificial neural networks (ANN), GP can be less sensitive to the amount of available data [30]. This is advantageous in animal science, where databases are usually small.

The appropriate fitting performance of the MP algorithm can be related to its extremely flexible structure, which allows it to represent a wide range of response surface shapes [31]. In this sense, when enough data are provided, the MP algorithm can adequately model curvatures, interactions, plateaus, and step functions. In our study, the ability of this algorithm to deal with different shapes of lactation curves compared to the Wood model is demonstrated by the better estimates of TPY, PY, and TPY. In addition, MP (as a type of ANN) does not require the standard assumptions of regression, such as the mutual independence of the true residual, the normal distribution, and the constant variance [31]. 

MP is actually a non-linear model; however, this algorithm has several advantages over non-linear regression models. Firstly, MP uses activation functions (e.g., sigmoid, arcus tangens, etc.) that are flexible and can be adapted to a wide range of circumstances [32]. Additionally, MP minimizes the sum of squares errors. This is performed using a gradient descent approach called backpropagation. This approach calculates by sweeping forward and backward through the network to update weights and bias parameters to make ANN predictions more accurate [33].

The M5 and M5P showed exact goodness of fit for the estimation of the characteristics of the lactation curve in dairy ewes. This is in agreement with Witten et al. [12], who indicated that the rules algorithm (M5) and its corresponding decision tree (M5P) produce exactly the same predictions. However, rule sets can be more confusing when decision trees suffer from replicated subtrees. Additionally, in multiclass situations, the coverage algorithm focuses on one class at a time, whereas the decision tree learner considers all classes [12]. In our study, this situation could be evident if lactations were classified by their shape (typical or atypical), number of lambs (primiparous and multiparous), and type of lambs (single or multiple). The analysis according to the mentioned class can be useful mainly for PY and TPY, which showed a high variability of their values (Table 2). Among the logarithms used and proposed in the current study, the RF is the one most widely used in the animal science field. This algorithm has been particularly useful in the solution of classification problems [34] and in genetic improvement programs [35,36]. Although the goodness of fit of RF was lower in our study compared to SMOreg, M5, and M5P, RF has the advantage of its interpretable architecture, which contrasts with other ML algorithms considered uninterpretable black boxes [35].

## 5. Conclusions

The result of the current study shows that the ML algorithms make it possible to adequately predict the characteristics of the lactation curve with a relatively small number of input data. In fact, five monthly records of milk production were sufficient to obtain adequate estimates of TMY, PY, and TPY using the ML algorithms, overcoming the predictive capacity of the Wood model. All the algorithms tested depicted good estimates with better performance shown by the SMOreg algorithm. The techniques used to generate the output attribute may explain the differences in fitting performance between the algorithms. In our case, the algorithms that focus on the generation of predictions of continuous variables are the most suitable (i.e., SMOreg). It is also important to point out that some ML algorithms provide an interpretable architecture that is useful for decision-making at the farm level (i.e., rules and decision trees), maximizing the use of the available information. In this sense, as a future trend, we are conducting a series of experiments with the aim of applying automatic rules extraction algorithms to provide a better interpretation and explanation of the dairy sheep lactation curves by means of obtaining linguistic rules. These actionable rules will facilitate the decision-making farm stage.

## Figures and Tables

**Figure 1 animals-13-02772-f001:**
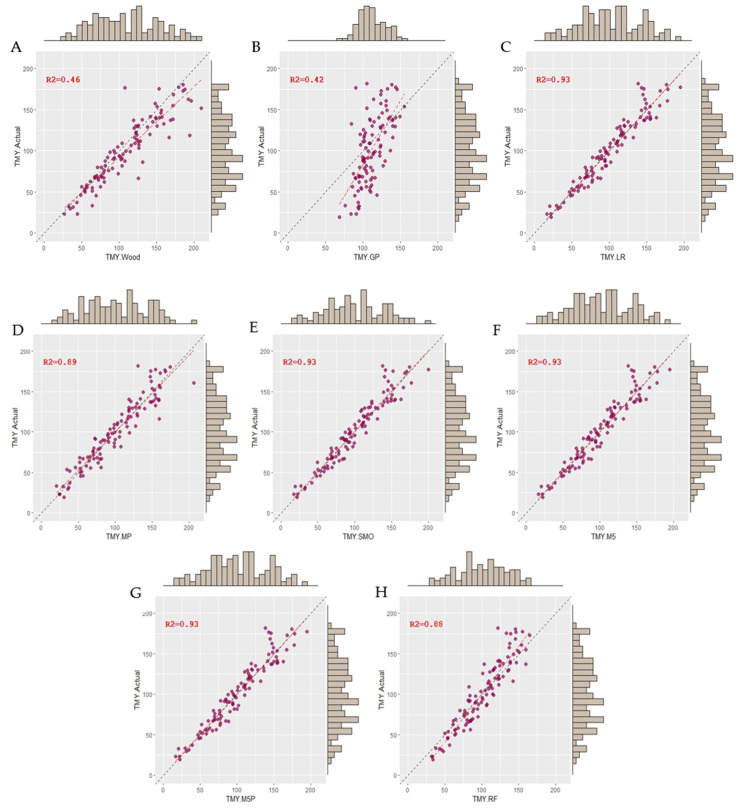
Scatter plots of predicted vs. observed values of the TMY. (**A**) Wood: gamma incomplete model, (**B**) GP: Gaussian process, (**C**) LR: linear regression, (**D**) MP: multi-layer perceptron, (**E**) SMOreg: sequential minimal optimization regression, (**F**) M5: M5 Rules, (**G**) M5P: model tree, and (**H**) RF: random forest algorithms.

**Figure 2 animals-13-02772-f002:**
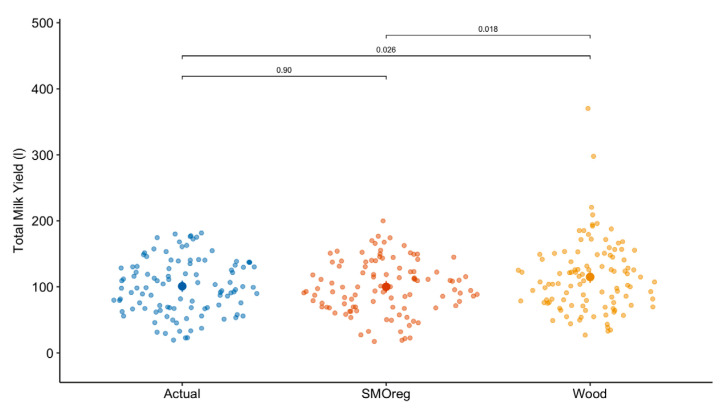
Actual and estimated values of total milk yield (TMY) of dairy sheep. In graphics: SMOreg, sequential minimal optimization regression; Wood, gamma incomplete model: *Y* = *at^b^* exp^−*ct*^.

**Table 1 animals-13-02772-t001:** Goodness of fit of the Wood model and ML algorithms for the characteristics of lactation TMY, PY, and TPY in dairy sheep.

	Wood	GP	LR	MP	SMOreg	M5	M5P	RF
Total Milk Yield (TMY)								
r	0.68	0.65	0.96	0.95	0.96	0.96	0.96	0.94
MAE	18.26	28.46	8.36	10.27	8.28	8.52	8.52	12.31
RMSE (l)	41.91	34.43	11.09	13.59	11.05	11.22	11.22	15.78
RAE (%)	52.53	81.59	23.96	29.44	23.72	24.42	24.42	35.29
RRSE (%)	101.48	83.10	26.77	32.79	26.67	27.08	27.08	38.09
Peak Yield (PY)								
r	−0.21	0.60	0.79	0.70	0.80	0.79	0.79	0.73
MAE	3.54	0.20	0.13	0.20	0.13	0.13	0.13	0.15
RMSE (l)	21.33	0.26	0.20	0.28	0.20	0.20	0.20	0.22
RAE (%)	1345.33	77.40	51.52	76.24	49.72	51.52	51.52	57.32
RRSE (%)	6409.03	82.03	61.80	85.93	60.81	61.80	61.80	67.95
Time of Peak Yield (TPY)								
r	0.47	0.66	0.65	0.54	0.69	0.65	0.65	0.63
MAE	19.89	13.33	12.09	17.36	10.05	12.06	12.06	12.88
RMSE (l)	29.12	17.66	17.97	22.51	17.68	17.92	17.92	18.10
RAE (%)	101.46	67.57	61.26	88.01	50.96	61.11	61.11	65.29
RRSE (%)	124.48	74.95	76.26	95.53	75.03	76.06	76.06	76.79

GP: Gaussian processes, LR: linear regression, MP: multi-layer perceptron, SMOreg: sequential minimal optimization regression, M5: M5Rules, M5P: M5P tree, and RF: random forest. r: coefficient of correlation, MAE: mean absolute error, RMSE: root mean squared error, RAE: relative absolute error, and RRSE: relative root mean squared error.

**Table 2 animals-13-02772-t002:** Mean and statistical differences between the actual lactation curve characteristics and estimated ones by the Wood model and SMOreg algorithm.

	Actual	Wood	SMOreg	SE	*p*-Value
Total milk yield (l)	101.02 ^a^	115.08 ^b^	100.2 ^a^	4.44	0.03
Peak yield (l)	0.97	4.29	0.93	1.19	0.07
Time to peak yield (days)	30	23	25	2.35	0.07

^a,b^ Within a row, means followed by a common superscript do not differ significantly (*p* < 0.05).

## Data Availability

The data presented in this study are available on request from the corresponding author.

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
