# Peer review of "Application of Machine Learning Algorithms to Describe the Characteristics of Dairy Sheep Lactation Curves"

_animals, 2023, doi:10.3390/ani13172772_

Round 1
Reviewer 1 Report
The authors evaluated the performances of six machine learning algorithms in predicting sheep lactations and used the Wood model for performance calibration. The evaluation is useful and the results are interesting. But, I do not read why the Wood model is the best candidate for performance calibration. The database is quite confusing. I cannot arrive at the values presented.
The article simply presents the works of feeding data into software and interprets the outcomes. I, as an engineer, would like to read some more scientific information about the evaluated machine learning frameworks, e.g. line 92., and necessary mathematical representations, e.g. line 147. Mathematical symbols shall be explained and abbreviations shall be spelled out at their first time of appearances.
Several performance metrics were evaluated but only the RMSE is used. Why?
Author Response
Dear reviewer,
We want to thank you for providing the opportunity to improve the manuscript according with your valuable suggestions.
Reviewer #1: The authors evaluated the performances of six machine learning algorithms in predicting sheep lactations and used the Wood model for performance calibration. The evaluation is useful and the results are interesting. But, I do not read why the Wood model is the best candidate for performance calibration.
AU: We thank you for your valuable comment. The following sentence has been added into the discussion section to clarify this point:
“The Wood model is the most widely used mathematical function to describe the lactation curve, probably due to its simple mathematical structure and partial biological interpretation of its parameters [2].”
The database is quite confusing.
AU: The M&M section has been improved to provide more information on the source of data and the structure of the milk recording system.
I cannot arrive at the values presented.
AU: Sorry, the authors did not fully understand this comment.
The article simply presents the works of feeding data into software and interprets the outcomes. I, as an engineer, would like to read some more scientific information about the evaluated machine learning frameworks, e.g. line 92.
AU: Thank you for your valuable comment. The main objective of the paper is to test the forecasting capacity of a set of Machine Learning (ML) algorithms, using a dataset related with the dairy sheep lactation curves, in comparison with the Wood Model. In this sense, we have provide in section 2.2 (Model Formulation) a brief description and functionality of the ML algorithms with the better forecasting results, including references to each of one, in order to obtain a detailed functionality information. Additionally, we have included a paragraph (between line 94 and 104) explaining the WEKA environment that we have used to perform the experiments. Therefore, we considered that, due to the fact of the paper focus, the provided information is enough.
and necessary mathematical representations, e.g. line 147.
AU: Thank you for your comment. The Wood model has been added.
Yt = atbe-ct (1)
Where Y is the milk production (l) at time t, a is the production at the beginning of lactation, b represents the ascending phase and c describes the descending phase of the lactation curve
Mathematical symbols shall be explained and abbreviations shall be spelled out at their first time of appearances.
AU: Done
Several performance metrics were evaluated but only the RMSE is used. Why?
AU: All goodness-of-fit criteria provide useful information about the performance of the mathematical approaches used in the current work. However, RMSE is probably the most widely used and reliable estimate to measure the predictive accuracy of a model (Tedechi 2006). Furthermore, the RMSE is a very useful metric to evaluate the performance of mathematical models as it provides the prediction error in the original scale of measurement and can evaluate different prediction procedures regardless of their mathematical structure or the procedure used to obtain the estimated values. Therefore, the values of RMSE have been emphasized throughout the manuscript.
Tedeschi, L. O. (2006). Assessment of the adequacy of mathematical models. Agricultural systems, 89(2-3), 225-247.
Reviewer 2 Report
General comments:
The paper present an interesting approach to model lactation curves in sheep, however there is information missing that makes difficult to read the paper.
Material and method section is incomplete, more detailed description is needed. Type of breed, days in milk, length of the lactation, average milk production, parities. Are the models done by each one of the parities? Why you used some classification models instead of prediction? What are you trying to classify?
In the Materias and Methods why you use “visual inspection” Line 84, instead of a statistical method. Also more description of the database is needed, what was the breed, the length of the lactation, parities, average of the different performance parameters.
The models were done for all the parties? Because parities 1 and usually 2 are different, could this impact your results?
Also you made reference to 5 records need to model the lactation curve, but I am not sure where does this number came from? This needs more explanation.
The discussion section is more like a literature review than a discussion of your results. I would suggest to edited it accordingly
Specifics:
Line 27: had instead of have
Line 37: delete predict after adequately
Line 39: what does interpretable architecture means?
Line 51: to take instead of for taking
Line 69: was to evaluate instead of is the evaluation
Line 158: were instead of was
Line 189: how does the actual values were calculated?
LIne 193: need the description
Line 202: only 1 decimal is needed - here and the rest of the document
Line 204: be consisten on how you write it: written with and without the hyphen
Line 206 delete had a and add a was after fit
Line 209: how was this was done? Include the description on M&M section
Line 223: based on the visual?
Line 273: than the instead of then
Line 279: Why classification algorithms if the study was a prediction?
Line 323: Lams or sheep? - you discuss the number of lambs and parity but there is no information in the paper about this- please add this on the M&M section
Line 335: 5 monthly records or 5 milk records? Again this is not clear where is coming from and please clarify if it is 5 records per lactation or 5 monthly records?
were added in the general review
Author Response
Dear reviewer,
We want to thank you for providing the opportunity to improve the manuscript according with your valuable suggestions.
Reviewer #2: The paper present an interesting approach to model lactation curves in sheep, however there is information missing that makes difficult to read the paper.
Material and method section is incomplete, more detailed description is needed. Type of breed, days in milk, length of the lactation, average milk production, parities. Are the models done by each one of the parities? Why you used some classification models instead of prediction? What are you trying to classify?
AU: Done
In the Materias and Methods why you use “visual inspection” Line 84, instead of a statistical method. Also more description of the database is needed, what was the breed, the length of the lactation, parities, average of the different performance parameters.
AU: Thank you very much for your valuable comments. The following sentences have been added:
“Peak yield and time of peak yield were determined by identifying the highest values on each lactation curve.”
“These records were obtained from 156 multiparous (second and third lambing) sheep lactations.”
“We analyzed the lactation curves of dairy crossbred ewes from the following breeds: East Friesian, Pelibuey, Suffolk and Black Belly.”
“The average lactation length was 237.4 days with and total milk yield of 102.0 l, peak yield of 0.97 l and a time to peak yield of 30 days.”
The models were done for all the parties? Because parities 1 and usually 2 are different, could this impact your results?
AU: Thank you for your valuable comments. Actually, the number of lambs was included in the initial analysis without affecting the goodness of fit.
Also you made reference to 5 records need to model the lactation curve, but I am not sure where does this number came from? This needs more explanation.
AU: We appreciate your comment. Five month records were used in the current study. Monthly milk yield recording is the scheme (A4) proposed by ICAR for recording milk production in dairy ewes. This information has been specified in the M&M section.
“A total of 1,186 monthly records were used according to the A4 method proposed by ICAR [9] as the standard method for recording milk production in dairy sheep.”
“Lactations with a minimum of five monthly records were selected, resulting in a database of 119 lactations”
“3) milk yield at first monthly recording (MP1; l/d); 4) milk yield at second monthly recording (MP2; l/d); 5) milk yield at third monthly recording (MP3; l/d); 6) milk yield at fourth monthly recording (MP4; l/d); 7) milk yield at fifth monthly recording (MP5; l/d).”
The discussion section is more like a literature review than a discussion of your results. I would suggest to edited it accordingly
AU: The aim of the current study is to evaluate the ability of ML algorithms to estimate lactation curve characteristics in dairy sheep. There is limited information about these analytical strategies applied to animal science and milk production. For this reason, a significant part of our discussion is focused on the characteristics, advantages and weaknesses of the proposed ML algorithms. This may be useful for readers interested in the application of new techniques to the analysis of data from biological processes. However, we have restructured the discussion section to highlight the findings of the current study.
Specifics:
Line 27: had instead of have
AU: Done
Line 37: delete predict after adequately
AU: Done
Line 39: what does interpretable architecture means?
AU: We appreciate your comment. In Machine Learning “interpretable” means that it is possible of being understood by humans on its own. M5 Model Tree, Random Forest and, M5Rules are interpretable due to the fact that are based in decision trees, therefore we are capable to follow the forecasting trajectory and it is possible to generate linguistic rules from the tree, that are very intuitive and easy to understood for the human expert domain, therefore, these rules provide more interpretable information in decision making stage.
Due to the current paper focus, we are only interested in the forecasting results; however, we are working in a new paper where we are interested to extract, directly from the data, interpretable and expressive rules that can explain in a better way the dairy sheep lactation curves. In this new paper we will present the rules extracted from the decision trees algorithms like M5, Random Forest, M5 Rules and other ML algorithms capable to extract linguistic rules from the data, for instance, LR-FIR [Castro et al., 2011], OSRE [Etchells et al., 2006] and others.
We have extended the conclusion section by including a more detailed explanation about the future works of the paper, in order to provide more explainable structures for farming decision-making.
T.A. Etchells, P.J.G. Lisboa, Orthogonal search-based rule extraction (OSRE) method for trained neural networks: a practical and efficient approach, IEEE Trans. Neural Netw. 17 (2) (2006) 374–384.
Castro, F., Nebot, A., Múgica, F. (2011). On the extraction of decision support rules from fuzzy predictive models. Applied Soft Computing, June, 2011, Vol. 11, issue 4, Pp. 3463-3475. ISSN: 1568-4946
Line 51: to take instead of for taking
AU: Done
Line 69: was to evaluate instead of is the evaluation
AU: Done
Line 158: were instead of was
AU: Done
Line 189: how does the actual values were calculated?
AU: This information is provided on lines 83-85 as follows:
Actual total milk yield was calculated from monthly records of milk production using the centering day method or Fleischmann's method [10]. Peak yield and time of peak yield were determined by visual inspection of lactation curves.
LIne 193: need the description
AU: Thank you very much for your valuable comments. The following sentence has been added:
Where parameter a had a value of 4.006 l, corresponding to average milk production at the beginning of lactation. The parameters b (0.055) and c (0.006) represent the ascending and descending phases of the lactation curve. The values of parameters b and c >0 indicate the probable presence of atypical curves.
Line 202: only 1 decimal is needed - here and the rest of the document
AU: Done
Line 204: be consisten on how you write it: written with and without the hyphen
AU: Thank you for your valuable comments. “goodness of fit” instead “goodness-of-fit”
Line 206 delete had a and add a was after fit
AU: Done
.
Line 209: how was this was done? Include the description on M&M section
AU: Thanks for your comment. The following information has been added to the M&M section:
Finally, the observed values (Y) and the estimated values (X) predicted by the Wood models and the ML algorithms were fitted using a linear regression model. The regression line and the predicted values were presented graphically in a scatter plot of predicted vs. observed values. In these scatterplots, the ordinate and abscissa have the same scale and a 45-degree line has been drawn to facilitate their interpretation. The accuracy of the estimates is represented by the distance of a given point from the 45-degree line.
Line 223: based on the visual?
AU: Thanks for your comment. According with Pineiro et al. (2008), scatter plots of predicted vs. observed values are one of the most common alternatives to evaluate model predictions. However, we share the concern about interpreting goodness of fit based on these plots alone. Therefore, the scatter plots are complemented by other goodness of fit criteria, which is in line with Pineiro et al. (2008) who pointed out that this approach should be complemented (not replaced) by other statistics that add important information for model evaluation.
Line 273: than the instead of then
AU: Done
Line 279: Why classification algorithms if the study was a prediction?
AU: Thanks for your comment. The sentence has been deleted.
Line 323: Lams or sheep? - you discuss the number of lambs and parity but there is no information in the paper about this- please add this on the M&M section
AU: We agree with the reviewer. Although we have now shown this information in the M&M section, the current study did not use the number and type of lambing to adjust the machine learning algorithms. We considered this to be a limitation of our study. Therefore, we have included this paragraph:
“In our study, this situation could be evident if lactations were classified by their shape (typical or atypical), number of lambs (primiparous and multiparous) and type of lambs (single or multiple). To carry out the analysis according to the mentioned class can be useful mainly for PY and TPY, which showed a high variability of their values (Table 2).”
Line 335: 5 monthly records or 5 milk records? Again this is not clear where is coming from and please clarify if it is 5 records per lactation or 5 monthly records?
AU: Thank you for your valuable comments. The current study used five monthly records. This information is specified in the M&M section.
Reviewer 3 Report
58: apart form the abstract it is the first time you use ML – give a full name and add abbreviation in brackets
Introduction – can you add more information on ML?
Material – how were sheep kept? Were all of the same system?
91: TMY, TPY – explain
level of English is good, some minor editing issues
Author Response
Dear reviewer,
We want to thank you for providing the opportunity to improve the manuscript according with your valuable suggestions.
58: apart form the abstract it is the first time you use ML – give a full name and add abbreviation in brackets
AU: Thank you for your valuable comment. ML term has been described.
Introduction – can you add more information on ML?
AU: Done. The following paragraph has been added.
“Mathematically, the shape of the lactation curve is defined by its characterizing parameters, such as total milk yield, peak yield and time to reach the peak yield [2]. Traditional statistical techniques such as linear regression, non-linear regression and random regression have been used to calculate the parameters of the lactation curve using empirical and mechanistic approaches [3]. In addition, in recent years, machine learning (ML) algorithms have emerged as powerful tools for predicting and modelling complex data. These algorithms have the ability to learn patterns and hidden relationships in large data sets.”
Material – how were sheep kept? Were all of the same system?
AU: The lactation curve data were obtained from a commercial dairy sheep farm. Therefore, all animals had the same management. The following information has been added to the M&M section to better describe the origin of the data:
“A total of 1,186 monthly records were used according to the A4 method proposed by ICAR [9] as the standard method for recording milk production in dairy sheep. These records were obtained from 156 multiparous (second and third lambing) sheep lactations located in Querétaro region, Mexico, with an average annual temperature of 17.3°C and an average annual rainfall of 485 mm. Lactations with a minimum of five monthly records were selected, resulting in a database of 119 lactations. We analyzed the lactation curves of dairy crossbred ewes from the following breeds: East Friesian, Pelibuey, Suffolk and Black Belly. The average lactation length was 237.4 days with a total milk yield of 102.0 l, peak yield of 0.97 l and a time to peak yield of 30 days.”
91: TMY, TPY – explain
AU: Done
Reviewer 4 Report
The ML algorithm has good predictive ability for the parameters of the lactation curve of dairy sheep, which facilitates the estimation of milk production in cases with incomplete records, providing significant value to dairy sheep farms. This paper evaluates the ability of seven machine learning algorithms to estimate the total milk production, peak production, and peak time during lactation in lactating sheep. Among them, the SMOreg algorithm showed the best performance. Finally, it was concluded that machine learning algorithms can be effectively applied to the feature description of lactation curves in dairy sheep. Although the topic is very interesting, It has certain limitations. There are some issues that should be addressed.
Specific comments:
1. In line 17:”using mathematical models through linear and non-linear regression”. The machine learning algorithm also includes linear regression and Nonlinear regression methods. What is the difference between the traditional method mentioned here and the machine learning algorithm mentioned below? I suggest you point out more clearly the differences between the algorithm you are using and traditional algorithms.
2. In line 41: I suggest you consider changing the keywords to better summarize the entire text.
3. In line 64, The sudden emergence of Artificial Neural Networks (ANN) may make people mistakenly think that you are about to use the ANN algorithm. From what I know, ANN is a machine learning algorithm and also a deep learning algorithm, but it is not compared with any ANN-related algorithms in this paper. In addition, why doesn’t this text use the updated ANN algorithm or deep learning algorithm? I believe adding a comparison of neural network algorithms in this manuscript will enrich the content of the paper.
4. In section 3, I suggest that when explaining the Results, we can first explain that the ML algorithm is superior to the Wood model, and then explain that the SMOreg algorithm is better among the ML algorithms, which makes the structure more logical. Moreover, in the manuscript, the SMOreg algorithm is mainly compared with the Wood model, and I suggest adding comparisons between the SMOreg algorithm and the other six algorithms.
5. Why is the entire paragraph explaining GP regression in lines 285-299, and what is its purpose? In addition, I also noticed that you have provided detailed explanations for other algorithms, but only did not explain the SMOreg algorithm. What is the reason for this?
6. In line 306, "ML (as a type of ANN)". To my knowledge, ANN (Artificial Neural Network) is a model in a machine learning algorithm, and you can check relevant information for confirmation.
7. Why should the algorithm proposed in this article be compared with the Wood model? The detection results of the Wood model did not perform well, could you please provide additional explanation?
8. The second paragraph of the introduction does not provide a detailed explanation of the drawbacks of traditional technology. The transition in line 58 with “However…” is too abrupt.
9. In the summary of section 2.4, which parameter is more important? It should be given emphasis in section 3.
Average
Author Response
Dear reviewer,
We want to thank you for providing the opportunity to improve the manuscript according with your valuable suggestions.
Reviewer 4: The ML algorithm has good predictive ability for the parameters of the lactation curve of dairy sheep, which facilitates the estimation of milk production in cases with incomplete records, providing significant value to dairy sheep farms. This paper evaluates the ability of seven machine learning algorithms to estimate the total milk production, peak production, and peak time during lactation in lactating sheep. Among them, the SMOreg algorithm showed the best performance. Finally, it was concluded that machine learning algorithms can be effectively applied to the feature description of lactation curves in dairy sheep. Although the topic is very interesting, It has certain limitations. There are some issues that should be addressed.
Specific comments:
In line 17:”using mathematical models through linear and non-linear regression”. The machine learning algorithm also includes linear regression and Nonlinear regression methods. What is the difference between the traditional method mentioned here and the machine learning algorithm mentioned below? I suggest you point out more clearly the differences between the algorithm you are using and traditional algorithms.
AU: Thank you for your valuable comment. Yes. Linear and nonlinear regression are considered as machine learning algorithms, however, to deal with complex data analysis, sometimes modern data analysis algorithms give better results compared to classical linear and nonlinear regression. Therefore, in this study, we have performed several experiments where the algorithms presented in section 2.2 achieved better prediction results. In order to avoid possible confusion, the paragraph has been rewritten according to your suggestions and modified as follows:
“Mathematically, the shape of the lactation curve is defined by its characterizing parameters, such as total milk yield, peak yield and time to peak yield [2]. Traditional statistical techniques such as linear regression, non-linear regression and random re-gression have been used to calculate the parameters of the lactation curve using empirical and mechanistic mathematical models [3]. In addition, in recent years, ML algorithms have emerged as powerful tools for predicting and modelling complex data. These al-gorithms have the ability to learn patterns and hidden relationships in large data sets.”
- In line 41: I suggest you consider changing the keywords to better summarize the entire text.
Thank you for your valuable comment. The keywords have been modified as follow:
Artificial intelligence; milk production; milk recording; sheep.
- In line 64, The sudden emergence of Artificial Neural Networks (ANN) may make people mistakenly think that you are about to use the ANN algorithm. From what I know, ANN is a machine learning algorithm and also a deep learning algorithm, but it is not compared with any ANN-related algorithms in this paper. In addition, why doesn’t this text use the updated ANN algorithm or deep learning algorithm? I believe adding a comparison of neural network algorithms in this manuscript will enrich the content of the paper.
AU: Thank you for your comment. In line 64 we presented one related research work, where ANN was used to estimate milk production in sheep, and indeed in the current paper we have applied a well-known ANN method called Multi Layer Perceptron (MLP), as has been described in section 2.2, between the lines 117 and 120 of the paper. As we expected MLP was chosen as one of the 7 algorithms with better forecasting results.
- In section 3, I suggest that when explaining the Results, we can first explain that the ML algorithm is superior to the Wood model, and then explain that the SMOreg algorithm is better among the ML algorithms, which makes the structure more logical. Moreover, in the manuscript, the SMOreg algorithm is mainly compared with the Wood model, and I suggest adding comparisons between the SMOreg algorithm and the other six algorithms.
AU: Thank you very much for your suggestions. Part of the results has been rewritten according to your suggestions. Also, the discussion section has been restructured based on your valuable comments, which has improved the fluency of our manuscript.
- Why is the entire paragraph explaining GP regression in lines 285-299, and what is its purpose? In addition, I also noticed that you have provided detailed explanations for other algorithms, but only did not explain the SMOreg algorithm. What is the reason for this?
AU: The aim of the current study is to evaluate the ability of ML algorithms to estimate lactation curve characteristics in dairy sheep. There is limited information about these analytical strategies applied to animal science and milk production. For this reason, a significant part of our discussion is focused on the characteristics, advantages and weaknesses of the proposed ML algorithms. This may be useful for readers interested in the application of new techniques to the analysis of data from biological processes. We agree with the reviewer that little information is provided to explain the superior performance of the SMOreg algorithm. Therefore, the following sentences have been added:
“In our study, the SMOreg algorithm showed the best performance for the estimation of TMY, PY and TPY, which can be related to the fact that SMOreg aims to find the best regression function by projecting into a high-dimensional space where linear optimization techniques are applied, and then projecting the resulting regression function back into the original low-dimensional feature space of the actual variables [25]. Also, this algorithm excels at predicting outcomes from databases containing noise and uncertain attributes due to its ability to normalize and separate data using a hyperplane as a decision boundary [3,4]. This is useful in biological problems such as milk production, where noisy data scenarios are common [5].”
- In line 306, "ML (as a type of ANN)". To my knowledge, ANN (Artificial Neural Network) is a model in a machine learning algorithm, and you can check relevant information for confirmation.
AU: We agree with the reviewer. There is an error in paragraphs 300-308 and 309-315; the term ML (machine learning) should be MP (multi-layer perceptron).
- Why should the algorithm proposed in this article be compared with the Wood model? The detection results of the Wood model did not perform well, could you please provide additional explanation?
AU: Thank you for your comment. As explained in the discussion, the Wood model is the most widely used mathematical function to describe the lactation curve, probably due to its simple mathematical structure and partial biological interpretation of its parameters
- The second paragraph of the introduction does not provide a detailed explanation of the drawbacks of traditional technology. The transition in line 58 with “However…” is too abrupt.
AU: Thank you for your valuable comment. We have rewritten the paragraph as follows:
“Mathematically, the shape of the lactation curve is defined by its characterizing parameters, such as total milk yield, peak yield and time to peak yield [2]. Traditional statistical techniques such as linear regression, non-linear regression and random re-gression have been used to calculate the parameters of the lactation curve using empirical and mechanistic mathematical models [3]. In addition, in recent years, ML algorithms have emerged as powerful tools for predicting and modelling complex data. These al-gorithms have the ability to learn patterns and hidden relationships in large data sets.”
- In the summary of section 2.4, which parameter is more important? It should be given emphasis in section 3.
AU: All the goodness-of-fit criteria provide useful information about the performance of the mathematical approaches used in the current work. However, RMSE is probably the most widely used and reliable estimate to measure the predictive accuracy of a model (Tedechi 2006). Also, the RMSE is a very useful metric to evaluate the performance of mathematical models due to provide the prediction error in the original scale of measurement and can evaluate different prediction procedures regardless of their mathematical structure or procedure to obtain the estimated values. Therefore, throughout the manuscript the values of RMSE were emphasized.
Tedeschi, L. O. (2006). Assessment of the adequacy of mathematical models. Agricultural systems, 89(2-3), 225-247.
Round 2
Reviewer 2 Report
The paper improve after the corrections.
As a general comment/ question do you plan to evaluate the ML models on first lactation data at some point?
Author Response
AU: Thank you for your comment. We are currently integrating a database with enough information to evaluate the use of ML on first lactation data.